# Environmental impact of single-use, reusable, and mixed trocar systems used for laparoscopic cholecystectomies

Linn Boberg[1]*, Jagdeep Singh[2], Agneta Montgomery[3], Peter Bentzer[1,4]

1 Department of Clinical Sciences Lund, Anesthesiology and Intensive Care, Lund University, Lund, Sweden,
2 Centre for Environmental and Climate Science, Faculty of Science, Lund University, Lund, Sweden,
3 Department of Clinical Sciences Malmö, Division of Surgery, Lund University, Malmö, Sweden,
4 Department of Anesthesia & Intensive Care Helsingborg Hospital, Helsingborg, Sweden

* linn.boberg@med.lu.se

## Abstract

### Introduction

Climate change is one of the 21st century's biggest public health issues and health care contributes up to 10% of the emissions of greenhouse gases in developed countries. About 15 million laparoscopic procedures are performed annually worldwide and single-use medical equipment is increasingly used during these procedures. Little is known about costs and environmental footprint of this change in practice.

### Methods

We employed Life Cycle Assessment method to evaluate and compare the environmental impacts of single-use, reusable, and mixed trocar systems used for laparoscopic cholecystectomies at three hospitals in southern Sweden. The environmental impacts were calculated using the IMPACT 2002+ method and a functional unit of 500 procedures. Monte Carlo simulations were used to estimate differences between trocar systems. Data are presented as medians and 2.5th to 97.5th percentiles. Financial costs were calculated using Life Cycle Costing.

### Results

The single-use system had a 182% higher impact on resources than the reusable system [difference: 5160 MJ primary (4400–5770)]. The single-use system had a 379% higher impact on climate change than the reusable system [difference: 446 kg $CO_2$eq (413–483)]. The single-use system had an 83% higher impact than the reusable system on ecosystem quality [difference: 79 PDF*$m^2$*yr (24–112)] and a 240% higher impact on human health [difference: $2.4 \times 10^{-4}$ DALY/person/yr ($2.2 \times 10^{-4}$-$2.6 \times 10^{-4}$)]. The mixed and single-use systems had a similar environmental impact. Differences between single-use and reusable trocars with regard to resource use and ecosystem quality were found to be sensitive to lower filling of machines in the sterilization process. For ecosystem quality the difference between the two were further sensitive to a 50% decrease in number of reuses, and to using a fossil

---

**Data Availability Statement:** All relevant data are accessible within the paper and in the Supplement file "S1 Appendix".

**Funding:** Linn Boberg (L.B) received funding from Lund University Agenda 2030 Graduate School and

Peter Bentzer (P.B.) received Swedish Government Funding (ALF) (grant number: 86626). The funders had no role in study design, data collection and analysis, decision to publish, or preparation of the manuscript.

**Competing interests:** The authors have declared that no competing interests exist.

fuel intensive electricity mix. Differences regarding effects on climate change and human health were robust in the sensitivity analyses. The reusable and mixed trocar systems were approximately half as expensive as the single-use systems (17360 € and 18560 € versus 37600 €, respectively).

## Conclusion

In the Swedish healthcare system the reusable trocar system offers a robust opportunity to reduce both the environmental impact and financial costs for laparoscopic surgery.

## Introduction

Climate change is considered as one of the major general health issues in the 21st century [1]. In western countries, the health sector has recently been suggested to contribute with 3-10% of the consumption-based greenhouse gas (GHG) emissions [2–4]. In addition, the health sector is responsible for the release of pollutants such as particulate matter and smog formation which have considerable effects on human health [5,6]. In Canada and the US such pollutats are estimated to cause an annual loss of 23 000 and 400 00 disease adjusted life years, respectively [3,6].

Surgical procedures are one of the most resource intensive activities in healthcare [7] and an analysis of the environmental impacts of surgical procedures using life-cycle assessment (LCA) methodology is suggested as a valuable tool to increase sustainability of health care by allowing comparisons of the environmental impact of products or processes [8]. Previous studies have compared the environmental impact of reusable and single-use surgical and anesthetic items such as laparotomy pads [9], scissors [10], gowns and drapes [11], sets of instruments for spinal surgery and vascular access [7,12] airway management items [13,14] and anesthetic drug trays [15]. In the majority of these studies the reusable alternative has been suggested to have a lower environmental impact than their single-use alternatives [9–11,13–15], but in some cases the opposite is true [7,12]. Thus, reusability of medical equipment does not always result in lower environmental impacts.

With shorter hospital stays, less pain and scaring, and better intra-operative visibility, the laparoscopic technique has gained worldwide popularity. Around 15 million laparoscopic procedures are performed annually worldwide [16] and one of the most common procedures is a laparoscopic cholecystectomy, removal of the gall bladder [17]. Improving the sustainability of laparoscopic surgery could enhance the overall sustainability of the health care system. The use of single-use trocars during laparoscopic procedures is increasing [16] despite their higher financial costs as compared to the reusable alternatives [17]. The mechanisms that drive this change are not yet understood, but are likely to include ease of use, perceived benefits with regard to sterility as well as personal preferences [8,17,18]. Further, little is known about the environmental impact of the increased use of single-use trocars in laparoscopic surgery. In a recent study, single-use trocars used for laparoscopic cholecystectomies were suggested to have a higher environmental impact than trocars containing both reusable and single-use parts, as assessed using Life Cycle Assessment (LCA) methodology in a UK setting [19]. Results of an LCA depend on assumptions such as the models of the trocars and local conditions with regard to for example waste management and energy sources, and further studies are needed to assess the external validity of these findings. Moreover, in order to provide a more robust basis for decision-making, there is a need to assess the precision of any differences in the environmental impacts.

The primary objective of this study was to evaluate and compare the environmental impacts of a single-use, a mixed, and a reusable trocar system for laparoscopic cholecystectomy. A secondary objective was to assess the financial costs of respective trocar system. For these purposes, we utilized LCA, and Life Cycle Costing (LCC) to evaluate the environmental impacts and economic costs of trocar systems used for laparoscopic cholecystectomies, respectively. To ensure the clinical relevance, we compared trocar systems in clinical use at three hospitals in southern Sweden. The hospitals use either only single-use, only reusable or a mix of reusable and single-use trocars. To assess the robustness of the results we performed multiple sensitivity analyses.

## Materials and methods

The study did not involve patients and no ethical permission was sought. To assess environmental impacts and financial costs LCA and LCC analyses were performed.

### Life cycle assessment

LCA is a quantitative method used to model a product's environmental impact, accounting for the impact from all phases of its life cycle such as material extraction, production, use, and waste scenarios. An LCA is divided into four steps (i) goal and scope, (ii) life cycle inventory of materials' and processes' input and outputs, (iii) life cycle impact assessment, where the inputs and outputs are sorted depending on their environmental or health impact, and (iv) interpretation of results, including sensitivity and uncertainty analyses [20–22].

**Goal and scope.**   We assessed the environmental impacts of three trocar systems used for laparoscopic cholecystectomies at three hospitals in Southern Sweden, each performing up to 350 such procedures annually. For one procedure four trocars are needed, two small and two large. A trocar consists of a cannula and an obturator (Fig 1).

One system consisted only of reusable trocars (Landskrona Hospital), one system was a mix of reusable and single-use trocars (Helsingborg Hospital) and one system consisted only of single-use trocars (Skåne University Hospital, Lund). A specification of the trocars in each system is provided in Table 1.

A cradle to grave approach was used, and the environmental impact from extraction of raw material, manufacturing, the use phase, and waste management was assessed in accordance with the international standard ISO 14044 guideline for conducting an LCA. Landskrona Hospital was used as the index hospital in the analysis. This means that all hospital related data, such as product storage information, the model of autoclave and washer disinfector, and waste treatment practice was collected from this institution. At Landskrona Hospital one large reusable trocar is used approximately 500 times during its ten-year lifetime and based on this, 500 laparoscopic cholecystectomies were used as the functional unit. As reusable trocars can break and have a shorter lifetime, or, in some cases be used more than 500 times two sensitivity analyses were performed where the functional unit was changed to 250 and 750 surgeries.

System boundaries are set to define which materials and processes to include in the assessment. We modelled the process and material flows within the system boundaries using Sima-Pro software version 9.1.1.1, PRe Sustainability B.V.. Data collection included reference to the database ecoinvent v3.6, ecoinvent Centre. System boundaries for both single-use and reusable trocars were set to include raw material and fiber production, for both the production of trocars and the production of their packaging. Waste scenarios and energy savings due to the recycling of packaging materials were included in the assessment. Similarly, transports from the manufacturer to the hospital and from hospital to waste management facilities were included (see figure A in S1 Appendix). An attributional approach was employed, meaning

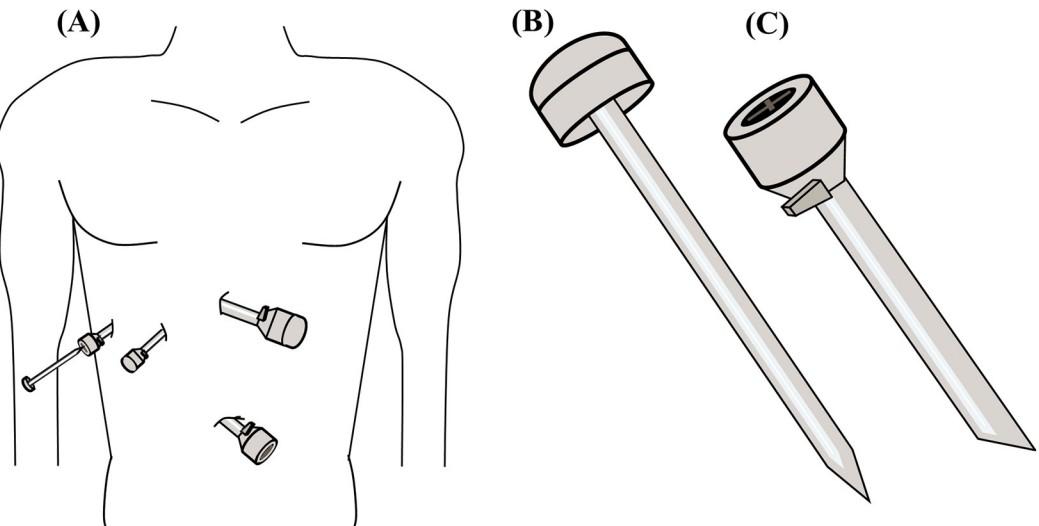

**Fig 1. Trocar.** (A) Placement of trocars for laparoscopic cholecystectomies. Each trocar consists of an (B) obturator and a (C) cannula.

that only environmental impact directly connected to the products' life cycles was accounted for [21]. Material and processes needed to produce the machines used in sterilization process (autoclave and washer-disinfector) fell outside the system boundaries, as recommended [22]. Further, for processes where the impact could be attributed to products other than the trocars, the allocated impact was adjusted according to the trocars' assumed contribution. In the sterilization process, trocars in the reusable system represented 2% of a fully loaded autoclave and about 8% of a fully loaded washer-disinfector, and inputs to SimaPro were allocated based on

**Table 1. Product summary.**

| System | Type of device | Country of origin | Uses / lifetime | Nr needed for one surgery | Nr needed for 500 surgeries |
|---|---|---|---|---|---|
| Single-use system (Lund) | Single-use trocar 5–12 mm | Ireland, Dublin | 1 | 2 cannulas/ 1 obturator | 1000 cannulas/ 500 obturators |
|  | Single-use trocar 5 mm | Ireland, Dublin | 1 | 2 cannulas/ 1 obturator | 1000 cannulas/ 500 obturators |
| Landskrona (Reusable system) | Reusable trocar 12 mm | Great Britain, Leeds | 500 | 2 cannulas/ 1 obturator | 2 cannulas/ 1 obturator |
|  | Reusable cannula 5 mm without stopcock | Great Britain, Leeds | 100 | 1 cannula | 5 cannulas (+ 500 membranes) |
|  | Reusable trocar 5 mm with stopcock | Great Britain, Leeds | 100 | 1 cannula/ 1 obturator | 5 cannulas/ 5 obturators (+ 500 membranes) |
| Helsingborg (Mixed system) | Single-use trocar 5–12 mm | Netherlands, Amsterdam | 1 | 1 cannula/ 1 obturator | 500 cannulas/ 500 obturators |
|  | Reusable trocar 10 mm | Germany, Frankfurt | 500 | 1 cannula/ 1 obturator | 1 cannula/ 1 obturator (+ 90 membranes) |
|  | Reusable trocar 5 mm | Germany, Frankfurt | 100 | 2 cannulas/ 1 obturator | 10 cannulas/ 5 obturators (+ 200 membranes) |

Summary of the trocars included in the respective reference flow (RF), including from which hospital the data is gathered (LU, Skåne University Hospital Lund; LA, Landskrona Hospital; HBG, Helsingborg Hospital.), type of trocar, country of origin, number of uses per lifetime for each trocar, number of each trocar (cannulas and obturators) needed to perform one laparoscopic cholecystectomy, and total number of each trocar (cannulas and obturators) needed for 500 such procedures.

the trocars share of a fully loaded machine. In the mixed system the reusable trocars represented 1.5% of a fully loaded autoclave and around 6% of a fully loaded washer-disinfector. To assess a situation in which the machines were not fully loaded we increased the allocation by two and five times the original allocation in sensitivity analyses.

Data describing the specifics of the trocars, processes at the hospital and the hospital's waste management system was collected using questionnaires sent to the practitioners, the trocar distributors or manufacturers, and the supplier of sterilization machines. See table A in S1 Appendix for the quality of the data.

**Life cycle inventory.** Input data for trocars in the reusable-, single-use and mixed systems are presented in tables B-E in S1 Appendix. Data on some of the plastic materials was neither provided by the manufacturer nor included in the ecoinvent v3.6 database. To assess the impact of our assumptions concerning these materials we performed two sensitivity analyses in which all plastic materials were assumed to be the common plastics polycarbonate or high-density polyethylene.

Transportation distances for the trocars from the manufacturer (see country of origin in Table 1) to Landskrona (Sweden) were estimated using Google Maps, assuming use of the fastest route. Transport by road was modelled as a lorry weighing 16–32 metric tons with a Euro Class 5 engine which represents a large share of the European transportation fleet [23]. Transport by boat was modelled as freight by sea on ferry. Transport from raw material supplier to manufacturer was not modeled as the raw material impacts include average global transport. One of the manufacturers had air freight as an alternative and we therefore performed a sensitivity analysis using air freight as main mode of transportation.

The individual trocar package for new trocars was modelled based on data from the manufacturer of the trocars in the reusable trocar system. The sterilization wrap, used in the sterilization process and for the storage of the reusable trocars in between surgeries was modelled based on information from the index hospital (see table E in S1 Appendix).

Based on information from Getinge AB, the manufacturer of the sterilization machines, the sterilization process was modelled with water from well (tap water), deionized water, detergent (alkylbenzene sulfonate), average wastewater treatment for Europe, and a Swedish, mainly renewable, electricity-mix consisting of 39% hydropower, 39% nuclear power, 12% wind, 10% thermal [24] (see table F in S1 Appendix). To assess the impact of the electricity-mix we performed two sensitivity analyses using either a largely coal dependent electricity mix as exemplified by the Polish electricity-mix consisting of 72% coal and oil, 16% natural gas, 10% renewable [25] or a European standard market electricity mix consisting of 46% coal and oil, 25% natural gas, 13% renewable energy, 12% nuclear, 3% hydropower [25].

At the end-of-life, all trocars were assumed to be incinerated whereas paper and plastics from instrument packaging and sterilization wraps were considered to be recycled.

**Life cycle impact assessment.** We estimated environmental impacts on fifteen midpoint impact categories: mineral extraction, non-renewable energy, global warming, aquatic eutrophication, aquatic acidification, land occupation, terrestrial acidification and nutrification, terrestrial ecotoxicity, aquatic ecotoxicity, respiratory organics, ozone layer depletion, ionizing radiation, respiratory inorganics, non-carcinogens, and carcinogens. The integrated downstream effect of these impacts was characterized into the four endpoint categories resources, climate change, ecosystem quality, and human health using the IMPACT 2002+ methodology to get a holistic understanding of the overall environmental impact of the different product systems [26]. The unit for the resource endpoint is MJ Primary, referring to the total amount of extracted non-renewable energy. The unit for the climate change endpoint is kg $CO_2$eq 100, referring to the climate effect of $CO_2$ emitted into the air over 100 years. The unit for ecosystem quality is PDF*$m^2$*yr, referring to potentially dissapeared fraction of species over a certain area during a certain time. The unit for

human health is DALY, referring to disability adjusted life years per person and year [27] (For an illustation of suggested damage pathways please see figure B in S1 Appendix).

**Statistics and uncertainty analysis.** The ecoinvent v3.6 database provides an uncertainty range for each data point. The uncertainty of a data point is, if possible, based on the variation in sample data. For a data point which is based on a single source, without information on uncertainties, the ecoinvent database has a simplified standard procedure to estimate the uncertainty of an impact [28]. Monte Carlo simulation, which uses randomly selected input data within the uncertainty range for each parameter in the model, was used to simulate the 2.5th and 97.5th percentiles (iterations = 1000) [29,30]. Monte Carlo simulations were performed using SimaPro software version 9.1.1.1. As suggested previously for Monte Carlo simulations inferential statistics were not used to compare the systems [30,31]. Instead, dependent (paired) simulation in the Monte Carlo runs was used to assess the certainty of differences between the products [30,32]. This means that the same sample of input data was used for shared processes in the different systems. Data are presented as median and the 2.5th to 97.5th percentiles. Differences between the systems for which the 2.5th to 97.5th percentiles did not cross 0 were considered to reflect true differences.

## Life cycle costing

We used conventional LCC to measure direct financial costs [33] on the number of products needed for 500 surgeries (Table 1). The analysis included all financial costs for the hospital to buy, use and dispose of the different options. For the reusable options water and energy use, cost for the procurement and service of the sterilization machines, as well as labor cost for handling and maintenance were included (see table G in S1 Appendix). Two additional analyses were performed to test the sensitivity of the economic analysis regarding allocation made in the sterilization process and number of reuses.

## Results

### Life cycle impact assessment

The single-use system had the highest impact on the majority of the midpoint categories (see Table 2). When the midpoint categories' downstream effect was characterized into the four endpoints, we found that the single-use trocar system's impact on resources was 182% higher than the reusable system's impact [median difference of 5160 MJ primary (4400–5770)]. The impact on resources did not differ between the single-use and mixed trocar systems (Fig 2). The single-use trocar system's impact on climate change was 379% higher than the reusable system's impact and 12% higher than the mixed system's impact [median difference of 446 kg $CO_2$eq (413–483) and 55 kg $CO_2$eq (25–87), respectively] (Fig 3). The single-use trocar system's impact on ecosystem quality was 83% higher than the reusable system's impact [median difference of 79 PDF*m$^2$*yr (24–112)]. The impact on ecosystem quality did not differ between the single-use and mixed trocar systems (Fig 4). The single-use trocar system's impact on human health was 240% higher than the reusable system's impact and 6% higher than the mixed system's impact [median difference of $2.4 \times 10^{-4}$ DALY/person/yr ($2.4 \times 10^{-4}$–$2.6 \times 10^{-4}$) and $1.8 \times 10^{-5}$ DALY/person/yr ($1.3 \times 10^{-7}$-$4 \times 10^{-5}$), respectively] (Fig 5).

The median environmental impact of each product system on the fifteen midpoint categories with unit specification, presented with 2.5th to 97.5th percentiles. In the shaded columns the median differences between two alternatives are presented with 2.5th to 97.5th percentiles. A true difference was assumed between two alternatives if the 2.5th to 97.5th percentile range is located either above or below 0.

**Table 2. Midpoint impact category results.**

| Impact category (unit) | Single-use (percentile) | Reusable (percentile) | Mixed (percentile) | Single-use minus Reusable (percentile) | Single-use minus Mixed (percentile) | Mixed minus Reusable (percentile) |
|---|---|---|---|---|---|---|
| Mineral extraction (MJ surplus) | 1.1 (-0.03 to 3.9) | 4 (3 to 6) | 3 (2 to 5) | - 3 (-4.6 to -0.7) | -1.9 (-3.5 to -0.1) | 1 (0.25 to 1.9) |
| Non-renewable energy (MJ primary) | 8010 (7630 to 8470) | 2840 (2410 to 3560) | 7550 (7170 to 8160) | 5170 (4410 to 5770) | 462 (-143 to 1000) | 4700 (4550 to 4840) |
| Global warming (kg $CO_2$ eq) | 565 (535 to 602) | 118 (105 to 135) | 507 (486 to 532) | 446 (413 to 483) | 55 (25 to 87) | 389 (372 to 409) |
| Aquatic eutrophication (kg $PO_4$ P-lim) | 0.05 (0.04 to 0.09) | 0.02 (0.017 to 0.043) | 0.04 (0.03 to 0.07) | 0.03 (0.02 to 0.05) | 0.008 (0.003 to 0.016) | 0.02 (0.01 to 0.03) |
| Aquatic acidification (kg $SO_2$ eq) | 1.7 (1.6 to 1.9) | 0.41 (0.38 to 0.45) | 1.5 (1.45 to 1.55) | 1.3 (1.2 to 1.5) | 0.22 (0.13 to 0.34) | 1.08 (1.05 to 1.12) |
| Land occupation ($m^2$org. arable) | 85 (74 to 99) | 19 (15 to 28) | 41 (35 to 52) | 66 (56 to 78) | 44 (38 to 53) | 22 (19 to 26) |
| Terrestrial ecotoxicity (kg TEG soil) | 8380 (3260 to 14000) | 8560 (2610 to 16400) | 10100 (5140 to 16900) | 274 (-6670 to 4000) | -1730 (-7350 to 756) | 1810 (-392 to 3450) |
| Ter. acid-/nutrification (kg $SO_2$ eq) | 6 (5.8 to 7.2) | 1.4 (1.3 to 1.6) | 5.3 (5.1 to 5.6) | 4.8 (4.4 to 5.7) | 0.9 (0.5 to 1.8) | 3.9 (3.8 to 4.1) |
| Aquatic ecotoxicity (kg TEG water) | 35900 (27800 to 48400) | 35000 (23000 to 65500) | 41000 (30300 to 64600) | 2160 (-23100 to 14400) | -4650 (-29300 to 4190) | 5650 (-1850 to 12900) |
| Respiratory organics (kg $C_2H_4$ eq) | 0.21 (0.19 to 0.22) | 0.039 (0.036 to 0.042) | 0.16 (0.16 to 0.17) | 0.17 (0.16 to 0.18) | 0.04 (0.03 to 0.05) | 0.125 (0.12 to 0.13) |
| Ozone layer depletion (kg CFC-11 eq) | $1.8 \times 10^{-5}$ ($1.2 \times 10^{-5}$ to $3.2 \times 10^{-5}$) | $1.2 \times 10^{-5}$ ($9.6 \times 10^{-6}$ to $1.6 \times 10^{-5}$) | $1.7 \times 10^{-5}$ ($1.3 \times 10^{-5}$ to $2.5 \times 10^{-5}$) | $5.4 \times 10^{-6}$ ($-4 \times 10^{-8}$ to $1.9 \times 10^{-5}$) | $1.1 \times 10^{-6}$ ($-2.2 \times 10^{-6}$ to $8.1 \times 10^{-6}$) | $4.2 \times 10^{-6}$ ($2.2 \times 10^{-6}$ to $8.6 \times 10^{-6}$) |
| Ionizing radiation (Bq C-14 eq) | -754 (-2220 to 124) | 5800 (3650 to 23900) | 4510 (2760 to 19800) | -6650 (-24700 to -4120) | -5600 (-20600 to -3380) | -1330 (-4630 to -793) |
| Respiratory inorganics (kg PM2.5 eq) | 0.37 (0.34 to 0.4) | 0.1 (0.09 to 0.11) | 0.35 (0.32 to 0.36) | 0.26 (0.24 to 0.29) | 0.02 (-0.001 to 0.04) | 0.25 (0.24 to 0.26) |
| Non-carcinogens (kg $C_2H_3Cl$ eq) | 8 (5 to 14) | 5 (4 to 10) | 9 (6 to 18) | 2.5 (-0.4 to 6) | -1.6 (-4.7 to 0.9) | 4 (2 to 9) |
| Carcinogens (kg $C_2H_3Cl$ eq) | 22 (21 to 24) | 4.1 (3.6 to 4.9) | 18 (17 to 19) | 18 (17 to 19) | 4.3 (3.6 to 4.9) | 13.4 (12.8 to 14.7) |

Unit description (Jolliet et al., 2003), eq = equivalents.

MJ Surplus = MJ additional energy needed for future extraction.

MJ Primary = MJ total primary non-renewable energy extracted.

kg $CO_2$eq 100 = kg carbon dioxide emitted into air over 100 years.

kg $PO_4$ P-lim = kg orthophosphate as phosphorus into water.

kg $SO_2$ eq = kg sulfur dioxide into air.

$m^2$org.arable = m2 organic arable land.

kg TEG soil = kg triethylene glycol into soil.

kg TEG water = kg triethylene glycol into water.

kg $C_2H_4$ = kg ethylene into air.

kg CFC-11 = kg nitrous oxide into air.

Bq C-14 = Bq radiocarbon/ carbon-14 into air.

kg PM2.5 = kg particles with Ø > 2.5 μm.

kg $C_2H_3Cl$ = kg chloroethylene into air.

## Contribution analysis

Each life cycle phase's contribution to the respective system's impact is reported in detail in Fig 6. Briefly, the production of trocars and their packaging contributed by 70–95% of the impact

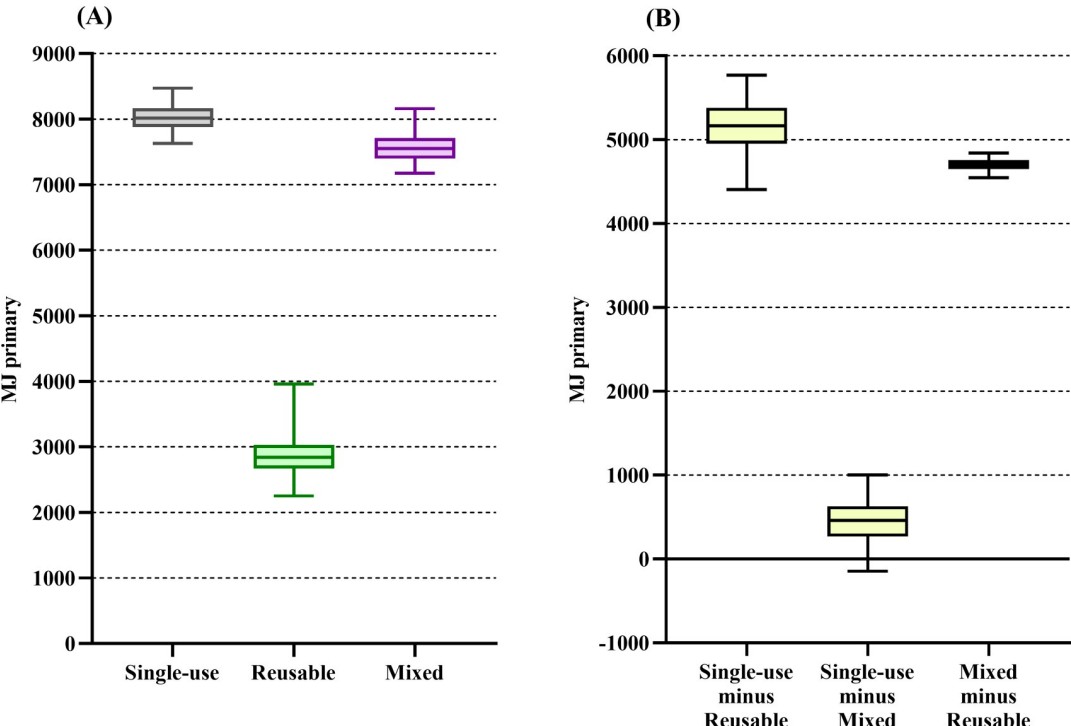

**Fig 2. Result for the resource endpoint category.** (A) Environmental impact of the reusable (green), mixed (purple) and single-use systems (grey) on the resource endpoint category. (B) The median differences between the respective systems. Data are presented as median and 2.5th to 97.5th percentiles. There is no difference between the two alternatives if the 2.5th to 97.5th percentile range cross 0 in panel B.

on the resource endpoint for all three product systems. Similarly, production and packaging were the largest contributors to climate change for all three product systems and represented 60–70% of the impact on this endpoint. The production phase represented 90% of the single-use system's impact, 65% of the mixed system's impact, and 35% of the reusable system's impact on the ecosystem quality endpoint. The sterilization process contributed to 35% of the mixed system's impact and 65% of the reusable system's impact on the ecosystem quality endpoint. For all three systems 75–90% of the impact on the human health endpoint came from the production phase. The similar environmental impact of the mixed and single-use trocar systems could be explained by the higher plastic weight of the single-use trocar used in the mixed system compared to the trocars used in the single-use system.

## Sensitivity analyses

The larger impact of the single-use system on climate change and on human health compared to the reusable systems remained in all sensitivity analyses. In contrast, the difference between these two systems regarding effects on the resource endpoint was no longer apparent when using a fossil heavy electricity mix in the sterilization process. Similarly, the larger impact of the single-use system on ecosystem quality was sensitive to a fossil heavy electricity mix and reduced number of reuses, as well as to an increased allocation in the sterilization process. With a five-fold allocation in the sterilization process the reusable system got a higher impact than the single-use system on this endpoint [median difference 143 PDF*m$^{2}$*yr (17–398)] (see figure C in S1 Appendix).

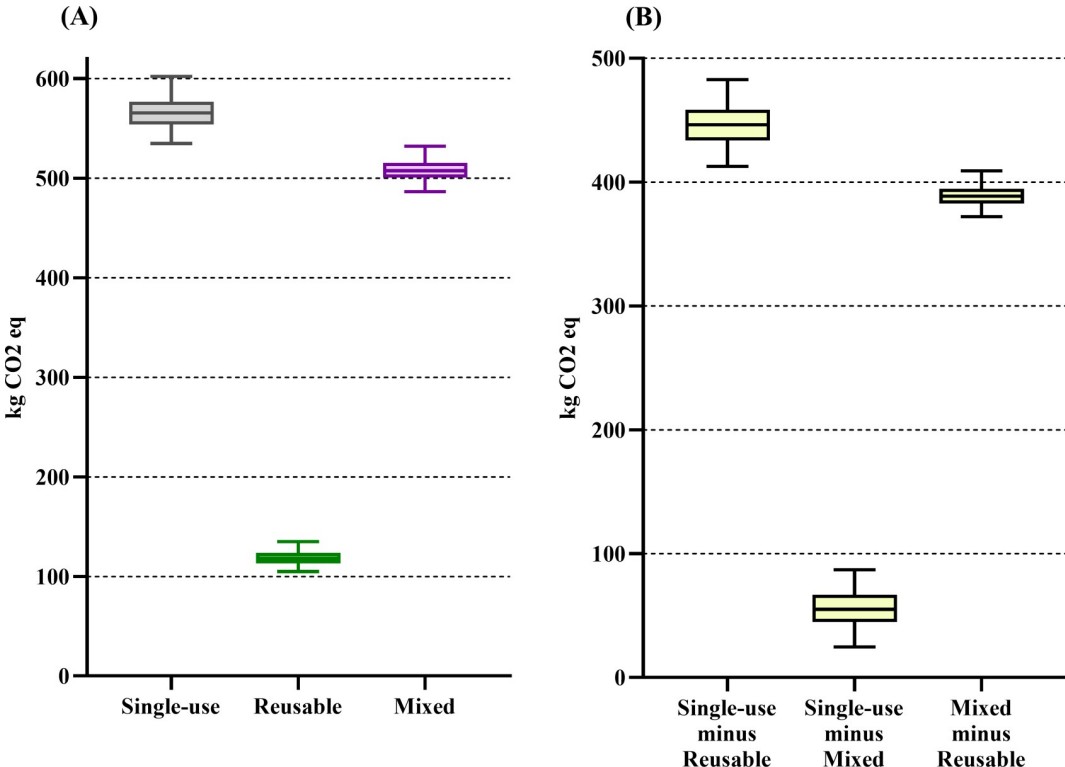

**Fig 3. Result for climate change endpoint category.** (A) Environmental impact of the reusable (green), mixed (purple) and single-use systems (grey) on the climate change endpoint category. (B) The median differences between the respective systems. Data are presented as median and 2.5th to 97.5th percentiles. There is no difference between the two alternatives if the 2.5th to 97.5th percentile range cross 0 in panel B.

The larger impact of the mixed system on climate change, human health and resources compared to the reusable system remained in all sensitivity analyses. The larger impact of the mixed system on ecosystem quality compared to the reusable system was sensitive to an increased allocation in the sterilization process (see figure D in S1 Appendix).

The single-use system´s larger impact on climate change compared to the mixed trocar system was sensitive to a five-fold allocation in the sterilization process and to changes in the electricity mix. When using a fossil heavy electricity mix the single-use system got a lower impact than the mixed system on this endpoint. Similarly, the larger impact of the single-use system on human health compared to the mixed trocar system was sensitive to an increased allocation in the sterilization process, to a change in electricity mix, to a decrease in the number of reuses, and to air freight as the main mode of transport. In the primary analysis there were no differences between the single-use and mixed systems' impacts on the resource and ecosystem quality endpoints. With a five-fold allocation in the sterilization process, the single-use system had a lower impact on both endpoints compared to the mixed system. When the functional unit was increased to 750 procedures the single-use system had a higher impact on both the resource and ecosystem quality endpoints compared to the mixed system (see figure E in S1 Appendix).

## Life cycle costing

The result of the LCC showed that the cost for the single-use trocar system was 59 660 euros, which is about twice as expensive as the reusable and mixed trocar systems (17 230 and 18 720

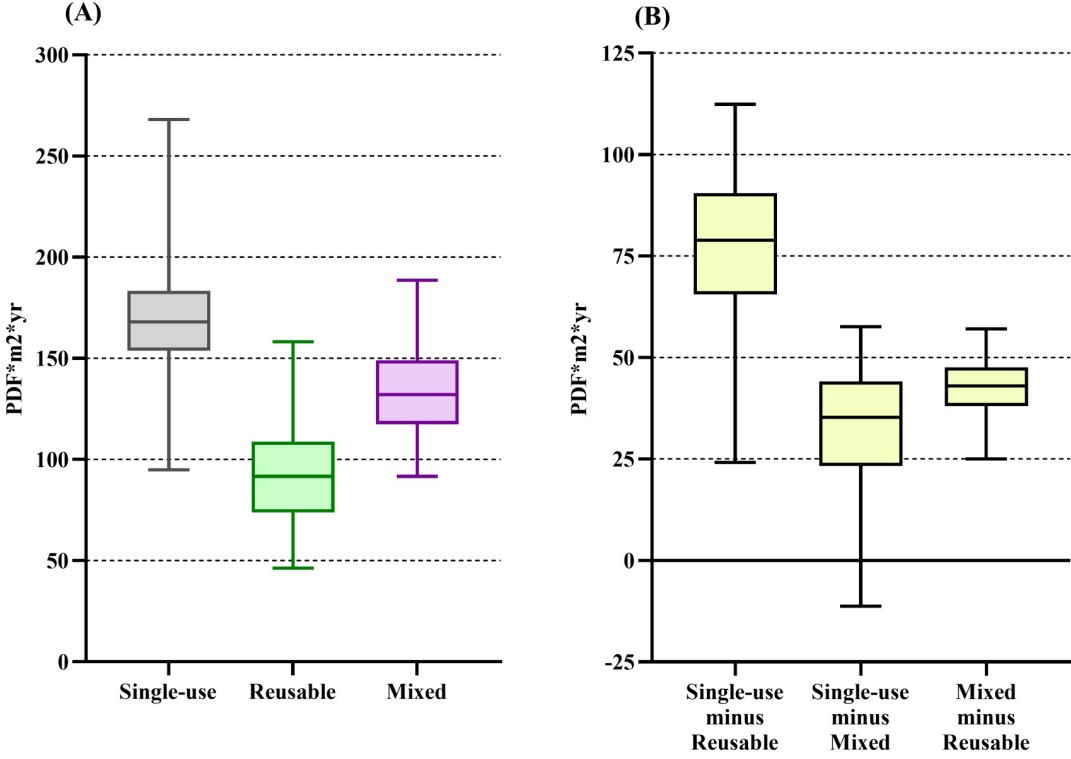

**Fig 4. Result for ecosystem quality endpoint category.** Environmental impact of the reusable (green), mixed (purple) and single-use systems (grey) on the ecosystem quality endpoint category. (B) The median differences between the respective systems. Data are presented as median and $2.5^{th}$ to $97.5^{th}$ percentiles. There is no difference between the two alternatives if the $2.5^{th}$ to $97.5^{th}$ percentile range cross 0 in panel B.

euros, respectively). The purchase cost was the major cost for all systems. For the single-use system, purchases represented over 99% of the total cost, for the mixed trocar system 73% (of which the major part could be referred to the purchase of single-use trocar) and for the reusable trocar system 63% of the total cost (with membranes being the largest expense). For the reusable and mixed trocar systems labor costs and the allocated costs of purchase and service of the autoclaves and washer-disinfectors were also large expenses. For the reusable system, the labor related costs represented 15% of the total cost, and allocated cost for the sterilization machines represented 12% of the total cost. For the mixed system, the labor related cost represented 11% of the total cost, and the allocated cost for the sterilization machines represented 8% of the total cost. Detailed specification of all costs is presented in table H in S1 Appendix.

We performed two sensitivity analyses concerning the number of possible reuses by changing the functional unit to 250 and 750 procedures. Further, we performed two sensitivity analyses for the mixed and reusable product system in which we increased the allocation by two and five times in the sterilization process. The single-use trocar system continued to be around twice as expensive as the reusable and mixed trocar systems, Table 3.

## Discussion

The results of this study show that the reusable trocar system has a lower environmental impact than the single-use and mixed systems. The sensitivity analyses suggest that the results are largely robust to assumptions with regard to materials, transports, and number of reuses. The financial cost of the reusable and mixed systems is lower than that of the single-use system.

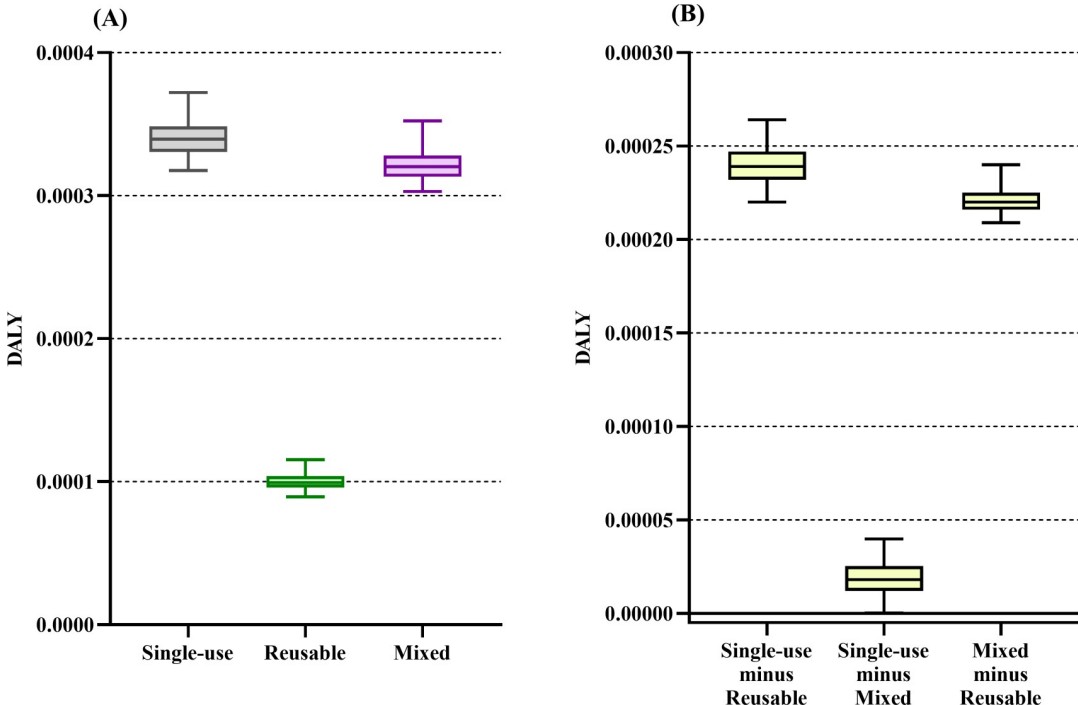

**Fig 5. Result for human health endpoint category.** Environmental impact of the reusable (green), mixed (purple) and single-use systems (grey) on the human health endpoint category. (B) The median differences between the respective systems. Data are presented as median with 95% and the 2.5th to 97.5th percentiles. There is no difference between the two alternatives if the 2.5th to 97.5th percentile range cross 0 in panel B.

The results of our environmental assessment align with results presented in a recent study suggesting that single-use laparoscopic trocar systems have a higher environmental impact than trocars largely consisting of reusable components [19]. Our analytical approach with uncertainty analyses together with multiple sensitivity analyses extends previous findings by providing an estimate of the precision and suggest that differences between the systems are robust. However, while the relative differences between the single-use systems and hybrid/reusable systems are similar in the two studies, the magnitude of the difference in absolute figures was up to 5 times as high in the study by Rizan and Bhutta [19]. This difference in results is most likely due to varied choice of system boundaries. The study by Rizan and Bhutta [19] included complementary instruments such as the storage tray, longer transportation distances, and a different source of electricity. The finding illustrates that modelling parameters are of great importance in assessment of the absolute effect of any given process. The different impact assessment methods could also contribute to the difference in results, as the ReCiPe and the IMPACT 20002+ methods compile data and characterizes midpoint and endpoint categories differently.

As mentioned in the introduction some hospitals use a mixture of reusable and single-use trocars in an attempt to save money and/or optimize performance. In our study, the rationale to use the single-use trocar in the mixed system was to reduce the risk of cannula displacement in the port where a lot of instruments are introduced and retracted during the procedure. Our results show that the mixed system, with only one single-use trocar, has almost the same environmental impact as the single-use system and suggest that the perceived improvement in function comes with considerable environmental impacts. This is dependent on the fact that

**Resources**

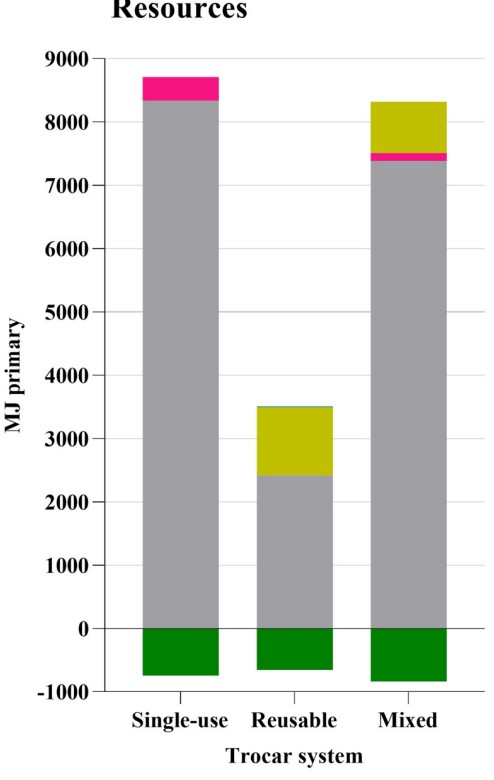

**Climate change**

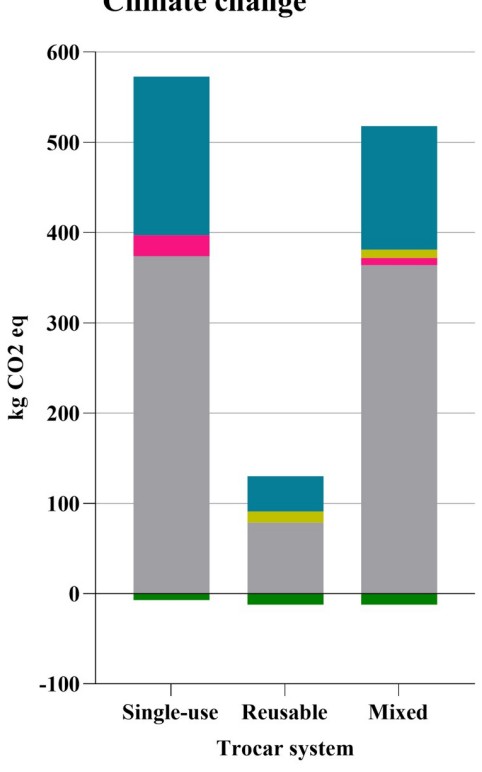

**Ecosystem quality**

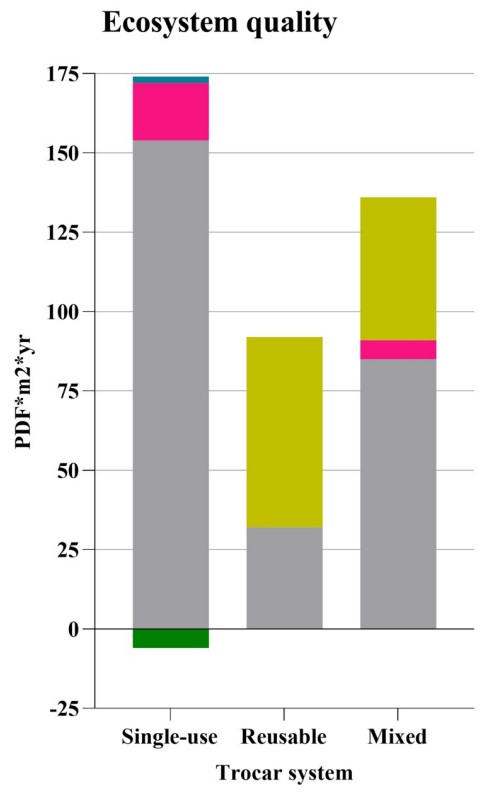

**Human health**

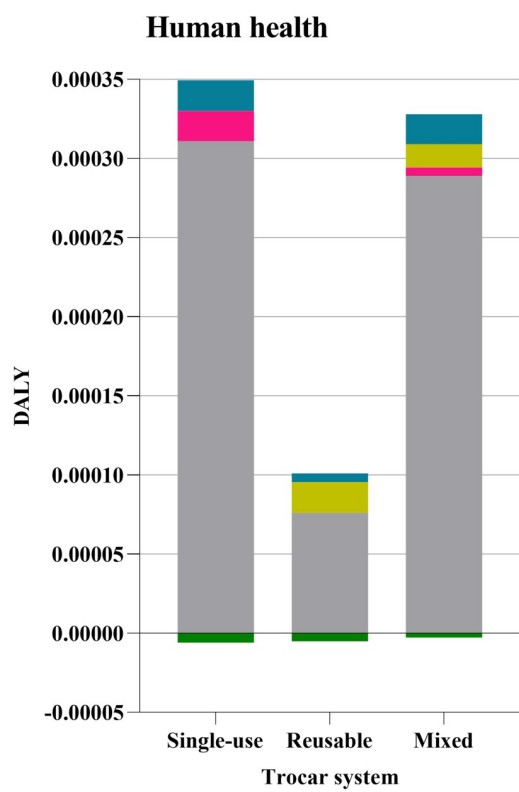

**Fig 6. Contribution analysis of endpoint category results.** The contribution from production (grey), sterilization (yellow), transport (pink), waste (blue) and recycling (green) to the product systems' impact on each endpoint category.

the single-use trocar used in the mixed alternative has a higher environmental impact than a corresponding trocar in the single-use system due to a higher plastic weight and highlight a potential to reduce environmental footprint of the mixed system.

The finding that single-use trocar systems have a higher environmental impact than reusable items align with a growing body of literature suggesting that the environmental impact of reusable products used perioperatively is lower than that of single-use alternatives. For example, reusable scissors [10], laryngeal masks [13], laryngoscope blades and handles [14], laparotomy pads [9], anesthetic drug trays [15], gowns and drapes [11], and sharps containers [34] have all been suggested to have a lower environmental impact than single-use alternatives. However, there are studies in which reusable instruments have been suggested to have a larger environmental impact than their single-use alternatives. Sets of instruments for spinal fusion surgery [7] and central venous catheter (CVC) insertion kits [12] with reusable components have been suggested to have a greater environmental impact than their single-use alternatives. In both studies this result could be explained by factors related to the sterilization process. Thus, in one study the reusable alternative included long transportation distances between each surgery and the sterilization facility [7], and in another study brown coal was used as the dominant energy source [12]. Although reusable options have a lower environmental footprint than their single-use counterpart in most settings, it is important to note that results cannot be generalized without careful consideration.

The contribution analysis identifies phases in a life cycle which are hotspots in terms of environmental impact. The production phase has been suggested as the major contributor to the impact on climate change of both reusable and single-use laparoscopic instruments [18]. Our contribution analysis extends these findings and identifies the production phase as the main source of the impact of reusable and mixed systems also on the resource and human health endpoints whereas the sterilization process has the largest influence on ecosystem quality. These findings are in contrast to studies showing that the sterilization process is the main contributor in all impact categories for reusable scissors [10,35]. The difference in results may in part be explained by the fact that there are single-use membranes in reusable trocars which have a large influence on the environmental impact of the reusable and mixed systems. Another explanation could be that the trocars in our original models were allocated a relatively small share of the total impact from the sterilization process. Taken together these findings suggest that improvements in the production process or development of reusable membranes, as well as improving wastewater treatment could be targeted to further reduce the environmental footprint of reusable trocars.

**Table 3. Life cycle cost analyses.**

| Scenario | Single-use trocar system | Reusable trocar system | Mixed trocar system |
|---|---|---|---|
| 500 procedures (primary analysis) | € 37 567 | € 17 359 | € 18 560 |
| 250 procedures | € 18 783 | € 10 643 | € 10 624 |
| 750 procedures | € 56 350 | € 24 076 | € 27 663 |
| 2-fold allocation, sterilization process | € 37 567 [a] | € 19 692 | € 22 400 |
| 5-fold allocation, sterilization process | € 37 567 [a] | € 26 690 | € 29 398 |

The cost of each product system in primary and sensitivity analyses.

[a] Based on the primary analysis since sterilization is not part of the single-use trocar system's life cycle.

In a recent review on cost and safety for single-use and reusable laparoscopic instruments it was concluded that reusable trocars appear to be economic advantageous, but uncertainties regarding hidden costs for reusable instrument such as cleaning, repair, labour cost and cost of initial capital investments were identified [8]. Our finding that the reusable trocar system is about 50% cheaper than the single-use system suggests that the cost for reusable trocars is lower even when these costs are considered in the analysis. Similar to previous studies from Germany and the UK we also found that the purchase of trocars is the main contributor to both the single and reusable systems financial costs [17,19]. We found that 40% of the total cost for the reusable systems comes from the purchase of single-use membranes and sterilization wraps. This highlights an area where cost savings could potentially be achieved. Taken together there is now increasing evidence to suggest that reusable trocars are cheaper than single-use alternatives.

About 800 000 laparoscopic cholecystectomies are performed annually in Europe [36]. We are not aware of any data on the fraction of single-use systems but assuming that single-use systems are used in half of these procedures our results suggest that a change to only reusable systems could save roughly 16 million Euro each year. Assuming a European electricity mix such a change would reduce the annual emissions of kg $CO_2$ equivalents with 40% or 360 tons, a reduction comparable with driving a medium sized passenger car 45 laps around the globe. It would also reduce the impact on resources with 33% or 4 136 000 MJ, equal to the annual energy needed for 311 average European households. Choosing reusable over single-use systems would reduce environmental impact on human health with 35% or 0.19 DALY/person/yr and reduce the impact on ecosystem quality with 23% or 61 000 PDF*m$^{2}$*yr, which is comparable to a biodiversity loss equal that of a 10% decrease in number of species per year on an area equal that of 9 soccer fields over a time period of 10 years.

It could be argued that there are differences between the systems with regard to clinical performance, which could be of importance when choosing between them. However, we are not aware of any data suggesting that single-use trocars offer an advantage relative the reusable trocars concerning patient important outcomes. The fact that all three systems in this study are real life systems used in different hospitals in close geographical proximity may also serve to support clinical equipoise.

## Limitations

One of the limitations of the study is that there is multiple single-use and reusable trocars on the market, the validity of our findings is uncertain for systems containing trocars other than those analyzed in this study. Similarly, we acknowledge that there are uncertainties in the modelling and in the input data that could have influenced our results. For example, we cannot exclude that the number of extra trocars opened during the procedure, use of spare parts (exceeding the normal change of membranes), or other instruments needed to perform the procedure may have varied between the systems. Due to the lack of data on regional and hospital specific wastewater treatment, we used an average European process instead which most likely has a different composition of pollutants and therefore might affect the accuracy of the results. In particular this may influence the water related midpoint categories and their effect on the ecosystem quality endpoint.

In the LCA, a second order energy analysis were made, meaning that materials and processes needed for production of capital goods [22] in the sterilization process was excluded from the system boundaries of the reusable and mixed trocar systems. The LCC did, however, include cost for the purchase and services of the machines used for sterilization. Consequently, there is a discrepancy between the LCA and LCC system boundaries. To get a more aligned

result, future studies could eliminate this discrepancy by conducting a LCA with system boundaries of the trocars' life cycles set to include materials and processes needed to manufacture the autoclave and washer-disinfector.

### Strengths

Strengths of the present study design include the fact that we compared trocar systems that are in clinical use and thus relevant from a clinical perspective. Another strength of present study design is the analytical approach with a comprehensive sensitivity analysis in which we assessed the robustness of our model to changes of the assumptions with regard to for example electricity mix, and the rigorous uncertainty analysis to assess the certainty of each of our results. We included several of the hidden costs for reusable laparoscopic instrument as previously identified [8] in the LCC analysis.

### Conclusion

For trocars there is a lack of evidence supporting clinical advantages and patient important outcomes for either single-use or reusable alternatives. Rather it is suggested that personal preferences and perceived economic benefits are the main drivers for the use of single-use items. We conclude that in a Swedish setting, the use of reusable trocar systems instead of single-use systems offer a robust opportunity to reduce both the environmental impacts and financial costs of laparoscopic cholecystectomies, without compromising quality of care.

### Supporting information

**S1 Appendix.**
(PDF)

## Author Contributions

**Conceptualization:** Linn Boberg, Peter Bentzer.

**Data curation:** Linn Boberg.

**Formal analysis:** Linn Boberg.

**Funding acquisition:** Peter Bentzer.

**Investigation:** Linn Boberg, Agneta Montgomery.

**Methodology:** Linn Boberg, Jagdeep Singh, Peter Bentzer.

**Project administration:** Linn Boberg.

**Supervision:** Peter Bentzer.

**Validation:** Linn Boberg, Jagdeep Singh, Agneta Montgomery, Peter Bentzer.

**Visualization:** Linn Boberg, Peter Bentzer.

**Writing – original draft:** Linn Boberg.

**Writing – review & editing:** Linn Boberg, Jagdeep Singh, Agneta Montgomery, Peter Bentzer.

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
