## [Decision Letter · Decision Letter 0]

4 May 2022

PONE-D-22-07043Environmental impact of single-use, reusable, and mixed trocar systems used for laparoscopic cholecystectomiesPLOS ONE

Dear Dr. Boberg,

Thank you for submitting your manuscript to PLOS ONE. After careful consideration, we feel that it has merit but does not fully meet PLOS ONE’s publication criteria as it currently stands. Therefore, we invite you to submit a revised version of the manuscript that addresses the points raised during the review process.

We look forward to receiving your revised manuscript.

Kind regards,

Huan Li

Academic Editor

PLOS ONE

Journal Requirements:

Reviewers' comments:

Reviewer's Responses to Questions

**Comments to the Author**

1. Is the manuscript technically sound, and do the data support the conclusions?

Reviewer #1: Yes

Reviewer #2: Yes

2. Has the statistical analysis been performed appropriately and rigorously? 

Reviewer #1: Yes

Reviewer #2: Yes

3. Have the authors made all data underlying the findings in their manuscript fully available?

Reviewer #1: Yes

Reviewer #2: Yes

4. Is the manuscript presented in an intelligible fashion and written in standard English?

Reviewer #1: Yes

Reviewer #2: Yes

5. Review Comments to the Author

Reviewer #1: This manuscript compared the environmental impact of single-use, reusable and mixed trocar system for laparoscopic cholecystetomies using Life Cycle Assessment. Furthermore, financial costs were also calculated using Life Cycle Costing. Totally, the submission is worthy of publication, in view of the detailed data and logical discussion. However, some parts of this paper still should be restructured and described, such as the Abstract and Conclusion. With an appropriate revision, this manuscript can be considered for publication.

Reviewer #2: General comments

This is a novel study that compares the cost and environmental impact of single-use, reusable, and mixed trocar systems used for laparoscopic cholecystectomies. It also quantifies the differences between single-use and reusable systems in terms of resources, climate change, ecosystem quality, and human health. However, certain details have to be performed out.

The introduction is rather brief, and certain concerns require more explanation. Although "surgical procedures are one of the most resource intensive activities in healthcare" and their environmental effects should be studied, there are various sorts of surgical procedures, and the author should explain why they studied laparoscopic cholecystectomy. Furthermore, related studies must be summarized and reviewed in order to identify research gaps and emphasize the objective and significance of this study.

The conclusion is far too brief. In the end, in addition to describing the new results, there should be some enlightenment or application.

The point is, I believe it is true that reuse can lessen environmental impact; nevertheless, whether reuse has any negative consequences, such as economic expenses and potential human health risks, the author must describe in thoroughly, otherwise readers may be misled.

Specific comments

Line 60-71, the authors should comment on the findings of the Ref. 12 (Environmental impact and life cycle financial cost of hybrid (reusable/single-use) instruments versus single-use equivalents in laparoscopic cholecystectomy), highlighting the contrasts between the two studies.

Line 104, Table 1. Product summary.

Line 188-200, the software (or processes) used to perform the Monte Carlo simulation, as well as the number of iterations required.

Line 213-216, delete.

Table 2, Global warming, CO2, subscript.

Table 3, Delete "Results,".

6. PLOS authors have the option to publish the peer review history of their article (what does this mean?). If published, this will include your full peer review and any attached files.

Reviewer #1: No

Reviewer #2: No

---

## [Author Response · Author response to Decision Letter 0]

1 Jul 2022

Dear Huan Li,

We want to thank the reviewers for the kind words and constructive comments. We have now revised the manuscript to address the concerns of the reviewers and hope that the manuscript can now be accepted for publication in PLOS ONE. Detailed responses to the reviewers are enclosed below.

Response to additional requirements:

 Reply: We have checked the file naming and it should now be in line with the style requirements of PLOS ONE. We have also run the figures through the PACE digital diagnostic tool to ensure that the figures meet PLOS requirements.

 Reply: It was our understanding that a financial disclosure was not supposed to be included in the manuscript as advised in the submission guidelines “Do not include funding sources in the Acknowledgments or anywhere else in the manuscript file. Funding information should only be entered in the financial disclosure section of the submission system”. We have now included a financial disclosure after the conclusion in the manuscript so that it matches the funding information given in the submission system and we hope that this is satisfactory.

3. Please review your reference list to ensure that it is complete and correct.

 Reply: No retracted paper has been cited. We have however rearranged the reference list after revising the manuscript and the following reference has been added:

- McGain, F., McAlister, S., McGavin, A., Story, D. A Life Cycle Assessment of Reusable and Single-Use Central Venous Catheter Insertion Kits, Anesthesia & Analgesia. 2012;114(5): 1073-1080.

Linn Boberg 

Reviewer: 1

We thank reviewer 1 for the kind words and constructive comments. Below is a response to each of the queries:

1. However, some parts of this paper still should be restructured and described, such as the Abstract and Conclusion. 

 Reply: This is a valid comment and we have added additional information in the abstract (introduction/conclusion) and the conclusion which we hope contributes to a clearer background and context for in which the results are valid.

Reviewer: 2

We thank reviewer 2 for the kind words and constructive comments. Below is a response to each of the queries:

1. The introduction is rather brief, and certain concerns require more explanation. Although "surgical procedures are one of the most resource intensive activities in healthcare" and their environmental effects should be studied, there are various sorts of surgical procedures, and the author should explain why they studied laparoscopic cholecystectomy. Furthermore, related studies must be summarized and reviewed in order to identify research gaps and emphasize the objective and significance of this study.

 Reply: This is valid remarks and we have now rewritten the introduction to clarify the context and rationale for choosing to assess different trocars used for laparoscopic cholecystectomies. We have also included a section with a summary of previous studies which has assessed or compared the environmental impact of reusable and single-use items. 

2. The conclusion is far too brief. In the end, in addition to describing the new results, there should be some enlightenment or application.

The point is, I believe it is true that reuse can lessen environmental impact; nevertheless, whether reuse has any negative consequences, such as economic expenses and potential human health risks, the author must describe in thoroughly, otherwise readers may be misled.

 Reply: This is a valid concern and we have changed the conclusion so that it is clear for which context the results are valid.

3. Line 60-71, the authors should comment on the findings of the Ref. 12 (Environmental impact and life cycle financial cost of hybrid (reusable/single-use) instruments versus single-use equivalents in laparoscopic cholecystectomy), highlighting the contrasts between the two studies.

 Reply: We have now provided a more detailed rationale for the need of additional studies in the field of laparoscopic surgery and emphasized the need for an assessment of the precision of any differences between the compared alternatives. Such an analysis was not presented in Ref 12. (now Ref. 19). We have now also added a sentence in the discussion in which we emphasize this analysis as an extension of the previous work presented in Ref 12. (now Ref. 19). 

4. Line 104, Table 1. Product summary.

 Reply: Thank you for this comment, this has now been corrected in our manuscript.

5. Line 188-200, the software (or processes) used to perform the Monte Carlo simulation, as well as the number of iterations required.

 Reply: Thank you for pointing this out, this has now been corrected in our manuscript.

6. Line 213-216, delete.

 Reply: Thank you for this suggestion, this has now been corrected in our manuscript.

7. Table 2, Global warming, CO2, subscript.

 Reply: Thank you for pointing this out, this has now been corrected in our manuscript.

8. Table 3, Delete "Results,".

 Reply: Thank you for this comment, this has now been corrected in our manuscript.

---

## [Editor Report · Decision Letter 1]

4 Jul 2022

Environmental impact of single-use, reusable, and mixed trocar systems used for laparoscopic cholecystectomies

PONE-D-22-07043R1

Dear Dr. Boberg,

We’re pleased to inform you that your manuscript has been judged scientifically suitable for publication and will be formally accepted for publication once it meets all outstanding technical requirements.

Kind regards,

Huan Li

Academic Editor

PLOS ONE
---

## [Editor Report · Acceptance letter]

7 Jul 2022

PONE-D-22-07043R1 

Environmental impact of single-use, reusable, and mixed trocar systems used for laparoscopic cholecystectomies 

Dear Dr. Boberg:

I'm pleased to inform you that your manuscript has been deemed suitable for publication in PLOS ONE. Congratulations! Your manuscript is now with our production department. 

Kind regards, 

on behalf of

Dr. Huan Li 

Academic Editor

PLOS ONE